# Long-Term Outcomes after Neonatal Hypoxic-Ischemic Encephalopathy in the Era of Therapeutic Hypothermia: A Longitudinal, Prospective, Multicenter Case-Control Study in Children without Overt Brain Damage

**DOI:** 10.3390/children8111076

**Published:** 2021-11-22

**Authors:** Elisa Cainelli, Luca Vedovelli, Emmanuele Mastretta, Dario Gregori, Agnese Suppiej, Patrizia Silvia Bisiacchi

**Affiliations:** 1Department of General Psychology, University of Padova, 35131 Padova, Italy; patrizia.bisiacchi@unipd.it; 2Unit of Biostatistics, Epidemiology and Public Health, Department of Cardiac, Thoracic, Vascular, and Public Health Sciences, University of Padova, 35121 Padova, Italy; luca.vedovelli@unipd.it (L.V.); dario.gregori@unipd.it (D.G.); 3S.C. Neonatologia, Ospedale S. Anna, Città della Salute e della Scienza di Torino, 10126 Torino, Italy; mastrettasartore@libero.it; 4Department of Medical Sciences, Section of Pediatrics, University of Ferrara, 44121 Ferrara, Italy; agnese.suppiej@unife.it; 5Padova Neuroscience Center (PNC), University of Padova, 35131 Padova, Italy

**Keywords:** perinatal asphyxia, cognition, neuropsychological, psychopathology, children

## Abstract

Background. Data on long-term outcomes in the era before therapeutic hypothermia (TH) showed a higher incidence of cognitive problems. Since the introduction of TH, data on its results are limited. Methods. Our sample population consisted of 40 children with a history of hypoxic-ischemic encephalopathy (HIE) treated with TH, with an average age of 6.25 years (range 5.5, 7.33), 24 (60%) males; and 33 peers with an average age of 8.8 years (6.08, 9.41), 17 (51%) males. Long-term follow-up data belong to two centers in Padova and Torino. We measured general intelligence (WPPSI-III or WISC-IV) and neuropsychological functioning (language, attention, memory, executive functions, social skills, visual motor abilities). We also administered questionnaires to their parents on the children’s psychopathological profiles and parental stress. Results. We found differences between groups in several cognitive and neuropsychological domains: intelligence, visuomotor skills, executive functions, and attention. Interestingly, IQ test results effectively differentiated between the groups (HIE vs. controls). Furthermore, the incidence of psychopathology appears to be significantly higher in children with HIE (35%) than in control peers (12%). Conclusions. Our study supports previous findings on a higher incidence of neuropsychological, cognitive, and psychopathological sequelae after HIE treated with TH. As hypothesized, TH does not appear to ameliorate the outcome after neonatal HIE in those children who survive without major sequelae.

## 1. Introduction

Great improvements in outcomes have been reported in children who experience neonatal hypoxic-ischemic encephalopathy (HIE) since the introduction of therapeutic hypothermia (TH). The first clinical randomized trials demonstrated that TH for full-term newborns with moderate to severe HIE significantly reduces mortality or neurodevelopmental disability at the age of 18−24 months [1,2]. However, normal neurodevelopmental outcomes in early childhood do not prevent cognitive and behavioral difficulties in late childhood and adolescence, because cognitive functions are not yet fully developed at this early age.

Long-term data (early and late childhood, adolescence) in the era before TH treatment showed a higher incidence of cognitive problems until adolescence (for a review, see [3]). Since the introduction of TH, data on its results have been limited: TH has been introduced too recently to adequately assess long-term effects, and several concerns are reasonable [1,2,4,5,6]. All but one study [4] extended randomized clinical trials to evaluate the efficacy and safety of TH in school-age children. Therefore, the participants in these studies were all children with a history of HIE, treated with or without TH. This type of comparison is optimal to highlight improvements and risks introduced by the new treatment, but not the gold standard to point out abnormal trajectories in neurodevelopment, which could emerge from comparisons with healthy controls. A healthy control group of typically developing children growing up in the same period may offer a more representative reflection of normal variation. Furthermore, group comparisons allow us to highlight subclinical vulnerabilities; that is, some borderline performances underlying weaknesses that could remain unchanged or worsen over time as more complex abilities emerge and the cumulative effects of several risk factors act synergistically.

To the best of our knowledge, the only study that compared children treated with TH with healthy controls during the school period found that children with HIE scored significantly lower in fine motor skills, executive functions, memory, and language [4]. As indirect confirmation, studies comparing children with HIE history who were treated with and without TH showed no significant neuropsychological differences, although differences clearly emerged for mortality and severe morbidity [1,2,5]. Given the established high incidence of long-term cognitive impairments before the era of TH usage, this result suggests a comparable incidence of neuropsychological difficulties despite TH treatment.

Ambiguity in prognosis is particularly evident in children with mild to moderate conditions. Most studies highlight predictive power based on neonatal parameters and examinations under severe conditions [7,8,9], but predicting functional disturbances, such as psychopathology and cognitive deficits, remains elusive. Furthermore, in clinical practice, infants with HIE who have normal or mildly abnormal short-term outcomes are considered low risk and discharged from the neonatal intensive care unit without dedicated long-term follow-up programs.

A clear understanding of the incidence and nature of impairments after HIE treatment with TH could increase clinicians’ and researchers’ attention to long-term follow-ups and the search of early biomarkers of risk. This can allow the identification and monitoring of the most vulnerable children and provide timely interventions if necessary. During the time elapsed between the insult and disclosure of impairments, the developmental window for therapeutic interventions may be lost. In early infancy, rehabilitation programs can still favor changes in brain circuitry when neural correlates underlying more complex adaptive abilities can still occur. Therefore, clear guidelines on follow-up and early biomarkers could be crucial to increasing the quality of life of children with HIE and reducing the burden on the mental health care system.

We compare the neurodevelopment of a group of children with HIE treated with TH to that of a control group of healthy peers, evaluating cognitive, neuropsychological, and psychopathological criteria. We hypothesize that despite the neuroprotection offered by TH, the early stressful condition could increase children’s risk of developing a wide range of neurodevelopmental disorders and subtle diseases, such as neuropsychological or learning impairments.

## 2. Materials and Methods

### 2.1. Participants

Our sample population consisted of 40 HIE patients with an average age of 6.25 years (range 5.5, 7.33), 24 (60%) males; and 33 peers with an average age of 8.8 years (6.08, 9.41), 17 (51%) males. All children fluently spoke Italian.

Our study focuses on a nested group of children selected from a larger prospective study on the prognostic role of evoked potentials and MRI in neonates with HIE who were eligible for TH from four Italian centers (Padova, Roma, Udine, Torino) from March 2013 to December 2015. Long-term follow-up data were collected for two centers, Padova and Torino. Selection for TH included (1) gestational age at birth ≥ 36 weeks; (2) any of the following: arterial umbilical cord or first blood gas analysis (within 1 postnatal hour) with pH 7.0, cord and base excess <12, 10-min Apgar score <5, or need for respiratory support at 10 min of life; and (3) moderate-to-severe encephalopathy within 6 h of birth.

The newborn exclusion criteria were suspected or known congenital malformations and inborn metabolism errors. TH was initiated as soon as possible after birth or at the time of referral from other hospitals and consisted of moderate whole-body hypothermia (target temperature 33–34 °C) for 72 h followed by a rewarming rate of approximately 0.5 °C/h. All patients received fentanyl infusion throughout TH to prevent discomfort and shivering (1–2 µg/kg/h, with boluses as needed).

For the long-term follow-up study, we selected children of at least 5 years of age with familial consent to participate in the study. Exclusion criteria for long-term follow-up were the presence of major impairments (neurosensory impairments, cerebral palsy, or epilepsy); diagnosis of congenital malformations after the neonatal period; inborn metabolism errors, genetic syndromes, or other medical comorbidities; certified intellectual disabilities; traumatic events or reported parental neglect; and invalidating parental pathologies that emerged during clinical follow-up.

The controls were recruited in a primary school. The project was presented and families were asked to join freely.

Only data from children who underwent the complete assessment battery were analyzed for this study. Parents gave their written informed consent for their children to participate.

### 2.2. Cognitive, Neuropsychological, and Psychopathological Assessment

A child psychologist and a child neurologist who were unaware of the medical status of the children assessed the patients. Children came to our clinic twice on successive days; several breaks were planned within each evaluation. Comfortable temperatures and a quiet environment were maintained throughout the evaluations. We also conducted a psychological interview with the parents of the children, who completed a questionnaire on their perceived level of stress from their parental roles and their children’s psychopathological profiles.

We measured general intelligence using the Wechsler Preschool and Primary Intelligence Scale III (WPPSI-III, [10]) test or the Wechsler Intelligence Scale for Children IV (WISC-IV, [11]). We used the naming test for language [12]. We used the visual and auditory attention tests of the NEPSYII [13] to measure attention. For memory, we used a word list and list recall [12], which evaluate learning and long-term verbal memory; and the Corsi test [12], which evaluates short-term visuospatial memory. To evaluate executive functions, we administered the coding test of the WISC-IV or WPPSI-III [10,11]; the semantic verbal fluency test [12], which evaluates the ability to access the lexicon through a categorical cue; and the Tower of London task [14], which evaluates planning. To assess social skills, we used the theory of mind A and B and affect recognition tests of the NEPSY-II [13]. Visual-motor abilities were measured using the Visual-Motor Integration Test [15].

All tests were corrected for age using published normative values standardized for the Italian population.

Parents completed the following psychopathological questionnaires:

Child Behavior Checklist (CBCL) [16]. The CBCL is a multiaxial, empirically based set of measures that assess the emotional, behavioral, and social problems of a child over the previous six months.

Conners Rating Scales—Revised (CRS-R) [17]. CRS-R reports parent ratings of child behaviors that involve problems in seven psychopathological areas: opposition, inattention, hyperactivity, anxiety, shyness, perfectionism, social problems, and psychosomatic issues.

Given the crucial role of other parental factors in determining psychological disturbances, parents also filled in the Short-Form Parent Stress Index (PSI-SF) [18], a standardized tool that produces scores of parental stress in four domains: parental distress, difficult child, dysfunctional parent–child interaction, and total stress. Patients with total stress clinically relevant elevations were excluded.

### 2.3. Statistical Analysis

Data are expressed as medians (interquartile ranges), or percentiles for the auditory attention task. Univariate analyses were performed with the non-parametric Wilcoxon rank sum test for continuous variables and Fisher’s exact test for categorical variables. No multiple comparison correction was applied, since we designed the experiment as a hypothesis generator [18,19]. Because the variables were on different scales, the data set was scaled and centered prior to analysis.

We calculated an index of psychopathological dysfunctions from clinical elevations of single psychological domains: we classified psychopathology as present if a patient attained an impaired score (>2 SD) in at least two psychological domains. This methodology provided dichotomous values and is useful to quantify a wide range of dysfunctions in a unitary measure. Fisher’s exact test was used for that dichotomous variable. We calculated positive predictive value, negative predictive value, sensitivity, specificity, and precision to evaluate the different psychopathologies present among children with HIE and controls.

Feature selection was implemented using a random forest algorithm by Boruta [20]. The Boruta algorithm aims to identify all the relevant predictors that impact the outcome of interest (in our case, belonging to the HIE or control group). It implements a random forest on an augmented set of covariates. Additional covariates, called shadow variables, are copies of the original variables obtained by permuting the observations and thus removing the eventual association with the outcome. For each explanatory variable, an importance measure is computed—that is, the Z score, which is the average improvement in the predictive performance of the random forest, with the explanatory variable divided by its standard deviation. Important predictors are those that show a Z-score greater than that observed for the variable with the highest Z-score among the shadow variables. This procedure is repeated until an importance measure is assigned to each predictor or until the maximum number of random forests is reached. We used the Boruta R package for analysis. Missing values were imputed only before the implementation of the Boruta algorithm with a robust random forest regression method with the R package {randomForestSRC} [21]. The *var.select* function of {randomForestSRC} was used to validate the results of the Boruta variable selection with the minimal depth (md) method and high conservativeness. All analyses were conducted in R v.4.1.1 [22].

## 3. Results

### 3.1. Descriptive Variables—Univariate Analysis

The differences between children with HIE and the control group in cognitive and neuropsychological performance are shown in Table 1.

Psychopathological results were not available for all children in the study: some parents did not respond to the questionnaires at all, and others did not complete all the questions. The results obtained are shown in Table 2.

Differences emerged in the psychopathological scores of children and controls with HIE: negative predictive value 0.88 (95% CI 0.75–1.00), positive predictive value 0.35 (95% CI 0.19–0.51), specificity 0.50 (95% CI 0.35–0.64), sensitivity 0.80 (95% CI 0.59–1.00), and overall accuracy 0.58 ( 95% CI 0.40–0.75), *p* = 0.04.

### 3.2. Feature Selection

Feature selection was implemented with two different algorithms based on random forest: Boruta, and minimal-depth variable selection with high conservativeness. Both algorithms independently confirmed IQ as the sole important variable for classifying the two groups of patients (See Figure 1). Results were consistent at different seeds of the random number generator.

## 4. Discussion

By examining cognitive, neuropsychological, and psychopathological profiles, we aimed to compare the neurodevelopment of children with a history of HIE treated with TH to that of a group of peers without preperinatal risk factors. As hypothesized, we found that children with HIE, even in the absence of severe disabilities, face a higher risk of developing a wide range of neuropsychological and psychopathological dysfunctions. We found differences in several cognitive and neuropsychological domains: intelligence, visuomotor skills, executive functions, and attention. Interestingly, the IQ test effectively differentiated between the groups (HIE vs. control group). Furthermore, the incidence of psychopathology appeared to be significantly higher in children with HIE compared to their control-group peers.

As in the study by Edmonds et al. [4], the most relevant difference in the neuropsychological profile was in the auditory attention test, an aspect of cognitive control important for school readiness [23] and long-term achievement [24]. Furthermore, we found differences in tasks evaluating visuomotor skills (Visual-Motor Integration Test and coding test). These abilities are typically impaired in children with a history of another pathological condition of the preperinatal period: preterm birth [25]. Despite the clinical differences between the disorders, mild-to-moderate conditions share common neurocognitive impairments that are considered disturbances of connectivity and are closely correlated with the integrity of the white matter [26].

However, IQ is the parameter that both differed significantly between groups and discriminated children with HIE from controls using very robust statistical procedures. IQ is a complex parameter obtained from numerous subtests that describe global cognitive functioning. However, total IQ was not interpretable in most of our patients due to the extreme variability between the indices (>1 SD). This means that the cognitive profile of children with HIE is not homogeneously lower than that of their peers, but that they show profiles characterized by deficits and preserved functions. This result is in line with our previous work on other preperinatal conditions [27,28,29]. As in those studies, we found high interindividual variability as demonstrated by highly variable scores in the performance of tasks or responses to the questionnaires. Developmental pathways may assume peculiar trajectories, resulting in high interindividual variability [26].

Unlike neuropsychological functioning, psychopathology has been very poorly explored in most pediatric neurological conditions. This is also the case for HIE. Although we did not explore the psychopathology extensively and our results are only preliminary, we found clinically relevant psychopathological symptoms in 35% of patients with HIE. The most frequent symptoms were oppositional behavior, attention problems, hyperactivity, somatic complaints, and anxiety. Although great attention has been paid to the neurobiology of some psychiatric disorders, historically, neurological correlates of psychopathology and psychological concerns have been ignored. Therefore, psychological symptoms in neurological diseases have been considered an effect of coexistence with a chronic disability. It is only recently that the view has changed [30]: for example, in the case of epilepsy, psychopathological and cognitive symptoms are now pivotally considered symptoms per se, like the seizures themselves [31]. We believe that the high percentage of psychopathological dysfunctions found among patients with HIE is likely a direct symptom of abnormalities in brain organization rather than a consequence of the acute stress experienced during the neonatal period. In this framework, cognitive and psychopathological disorders among our patients could be considered two different manifestations of the same underlying neurobiological vulnerability. However, only comparison with other similar pathological conditions without cerebral involvement could disentangle this ambiguity.

In HIE, the underlying pathophysiological mechanisms of functional impairments are not completely understood, especially for mild-to-moderate cases. Neuroimaging studies have described two macroscopic patterns of hypoxic-ischemic injury: deep gray matter damage after severe acute asphyxia (e.g., to the thalami, basal ganglia, or brain stem), and injury to paracentral “watershed” brain regions after prolonged partial asphyxia [32,33]. The first pattern has been associated with the most severe outcomes. However, there are no structural abnormalities in a high percentage of cases, as revealed by MRI and histochemical investigation. Animal studies indicate that cognitive and behavioral deficits may result from abnormal wiring of neural networks during development [34]. More specifically, neonatal hypoxic-ischemic injury is believed to disrupt large-scale functional pathways between the prefrontal cortex and the hippocampus; this is related to cognition [35] even when macroscopic morphological changes are minor or moderate. Since substantial improvements in neonatal care in recent years have decreased mortality and severe morbidity, subtle neuropsychological and psychopathological sequelae have become a challenge in patients with HIE.

A better understanding of the pathophysiology of hypoxic-ischemic injury in human neonates may directly improve the care of patients born with neonatal HIE. An interesting side effect would be to provide new insight into the understanding of early network development. As stated in [36], adverse influences early in development—particularly during intrauterine life and often due to oxidant injury—can increase the risk of the condition in adulthood. Decades of research confirmed that exposure to intrauterine adversity places children and adults at elevated risk of developing cognitive, social, emotional, and health problems [36,37], but pathophysiology remains elusive. Mechanisms involved in hypoxic-ischemic injury and cerebral plasticity may provide insight into fetal health and pathology programming [38].

This study has some limitations. Our sample size remains small; therefore, the results may not be completely generalizable. In particular, the sample of control children is too small to fully represent normality in cognitive, neuropsychological, or psychopathological domains, especially if one considers that the age group presents great interindividual variability. Furthermore, we did not obtain the scores of the questionnaires from all parents of the sample; in particular, the lack of control parent reports are about 24% of the entire sample of controls. However, to reduce possible bias, healthy children were not recruited based on the absence of current neuropsychological dysfunctions, but on the absence of preperinatal insults. Therefore, our healthy group reflects general population trends and the real occurrence of cognitive problems in children without known risk factors; most importantly, the differences between the groups are not due to an artificial superfunctioning group of peers. Furthermore, we controlled for some other influencing variables such as parental stress. We hope to replicate the study with a greater number of patients, perhaps comparing them to a group of children with another pathological condition. This process could better highlight some specificities of HIE.

Unfortunately, our group of children was too young to allow for a comprehensive evaluation of executive functions. Executive functions are a broad but important construct that generally refers to the processes involved in conscious control of thought and action, supported by a complex interaction between the frontal and subcortical circuits of the cerebral areas [39], which could also explain the broad spectrum of neuropsychiatric phenomenology [40]. In-depth executive evaluation of children with HIE could be a recommended future line of research. Regarding the in-depth assessment of executive functions, our patients were also too young for a complete evaluation of psychopathology. Our data are based on parents’ reports, but it is known that reports about psychological symptoms may differ significantly between children and parents. We think that these results have to be corroborated by following the children longitudinally. As children reach an older age, we believe it will be necessary to perform complete psychopathological examinations consisting of questionnaires and a clinical interview performed by a trained psychotherapist.

## 5. Conclusions

In conclusion, our study supports previous findings on a higher incidence of neuropsychological, cognitive, and psychopathological sequelae after HIE in children treated with TH. As hypothesized, TH does not appear to ameliorate the outcome after neonatal HIE among children who survive without major sequelae. This points to the importance of searching for early biomarkers of risk to improve prognosis and allow for prompt rehabilitation intervention.

## Figures and Tables

**Figure 1 children-08-01076-f001:**
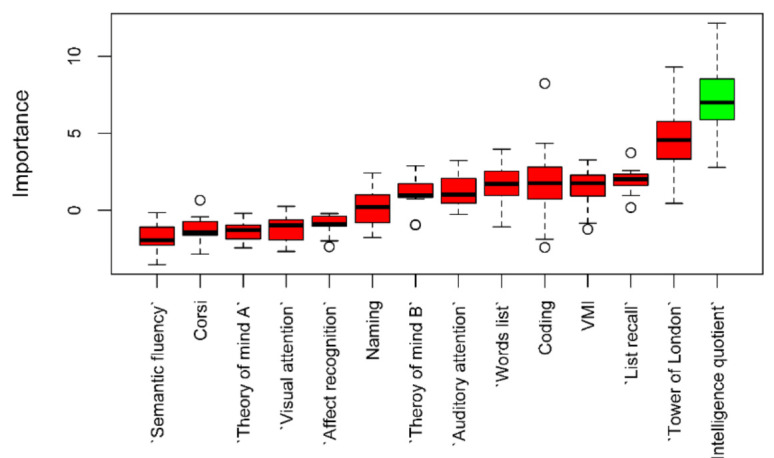
Boruta feature selection results. Green bar is IQ, the variable confirmed as important to distinguish the two groups of patients. Red bars represent the other variables included in the model but not confirmed as important. Dots in the upper or lower parts of the boxplots are outliers defined according to Tukey.

**Table 1 children-08-01076-t001:** Group characteristics by neuropsychological tests.

Characteristics	Controls, *N* = 33 ^1^	HIE, *N* = 40 ^1^	*p*-Value ^2^	Difference (95%CI)
Intelligence Quotient	105 (100, 115)	100 (87, 110)	0.031	8.7 (1.5, 16)
Coding	10.0 (8.0, 13.0)	8.0 (6.8, 10.0)	0.024	1.6 (−0.04, 3.2)
Semantic Fluency	−0.12 (−0.66, 0.63)	−0.25 (−1.06, 0.50)	0.4	0.23 (−0.28, 0.74)
Naming	0.12 (−0.27, 0.71)	0.00 (−1.04, 0.50)	0.3	0.33 (−0.13, 0.79)
Words list	0.19 (−0.15, 1.02)	0.22 (−1.02, 0.81)	0.4	0.50 (−0.12, 1.1)
Recall list	0.81 (0.00, 1.27)	0.14 (−0.57, 0.89)	0.052	0.53 (-0.01, 1.0)
Corsi	0.25 (−0.27, 0.78)	0.12 (−0.27, 1.11)	>0.9	−0.03 (−0.50, 0.44)
Visual-Motor integr.	13.00 (11.00, 15.00)	10.00 (9.00, 13.00)	0.016	1.9 (0.53, 3.3)
Tower of London	0.18 (−0.99, 0.66)	−0.72 (−1.46, 0.30)	0.079	0.67 (−0.02, 1.4)
Visual attention	11.00 (10.00, 12.00)	11.00 (9.00, 12.00)	0.3	1.0 (−0.36, 2.3)
Auditory Attention			0.004	
<2°	0 (0%)	1 (2.5%)		
2–5°	2 (6.1%)	2 (5.0%)		
6–10°	0 (0%)	6 (15%)		
11–25°	26 (79%)	18 (45%)		
26–50°	2 (6.1%)	11 (28%)		
51–75°	3 (9.1%)	2 (5.0%)		
>76°	0 (0%)	0 (0%)		
Affect recognition	10.0 (9.0, 11.0)	9.5 (7.0, 11.8)	0.5	0.63 (−0.87, 2.1)
Theory of Mind A	0.07 (−0.82, 0.68)	0.21 (−0.77, 0.94)	0.5	−0.12 (−0.66, 0.42)
Theory of mind B	0.12 (−0.36, 0.49)	−0.04 (−0.77, 0.34)	0.3	0.32 (−0.15, 0.79)

Legend: HIE: hypoxic-ischemic encephalopathy; ^1^ Median (interquartile range, IQR); n (%); ^2^ Wilcoxon rank sum test; Fisher’s exact test.

**Table 2 children-08-01076-t002:** Cross-tabulation of children and controls with HIE with present or absent psychopathology.

Psychopathology	HIE	Controls	Total
Present	12	3	15
Absent	22	22	44
Total	36	25	

## Data Availability

Data supporting the findings of this study are available from the corresponding author on a reasonable request.

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
