# Peer review of "Long-Term Outcomes after Neonatal Hypoxic-Ischemic Encephalopathy in the Era of Therapeutic Hypothermia: A Longitudinal, Prospective, Multicenter Case-Control Study in Children without Overt Brain Damage"

_children, 2021, doi:10.3390/children8111076_

Round 1

Reviewer 1 Report

The authors submitted a manuscript regarding a challenging topic, namely long term neurodevelopment outcome under several domains in HIE children treated with therapeutic hypothermia.

The authors investigated a group of 40 children who experienced a moderate/severe HIE promptly treated with HIE and compared them with 33 healthy controls.

Although the topic is undoubtedly interesting the reviewer believes several major issues in the paper needs to be addressed.

Introduction:

The entire section must be extensively revised as English is very poor and it is hard to read. 

Line 42: the reviewer suggests to indicate the duration of the follow-up from ref [3].

Methods:

The control group is definitely too small in order to fully represent 'normality' in cognitive, neuropsychological and psychopathological domains in an age ranging from 6 and 9 years. This vulnus deserve to be better addressed in the limitations paragraph.  

line 91: do the authors mean at least 36 week of age?

As all the children fluently spoke the Italian language it needs to be clarified in the text if all the tests performed (line 121-130) were actually validated in their nativa language.

Results:

In the reviewer opinion psychopathological results should be removed for two reasons:

1)comparison lacks in statistical significance due to the even smaller  control group sample;

2)results are based on parental questionnaires. Their subjective point of view should not be taken into account in building a prognostic/pathological tool.

The reviewer believes all the psychopathological analysis should then be removed from the main text and submitted as a supplementary material.

The discussion section need to be restructured accordingly in light of the revision.

A particular attention should be devoted to the limitations paragraph. The reviewer believes the sentence 'Therefore, our healthy group reflects general population trends and the real occurrence of cognitive problems in children without known risk factors' (line 270-271) cannot be stated with such a limited control group.

The results proposed, although interesting, are preliminary and need to the be extensively investigated on higher numbers and  challenged with a pathological group secondary to different causes than HIE.

Author Response

Reviewer 1

The authors submitted a manuscript regarding a challenging topic, namely long term neurodevelopment outcome under several domains in HIE children treated with therapeutic hypothermia.

The authors investigated a group of 40 children who experienced a moderate/severe HIE promptly treated with HIE and compared them with 33 healthy controls.

Although the topic is undoubtedly interesting the reviewer believes several major issues in the paper needs to be addressed.

 Introduction:

The entire section must be extensively revised as English is very poor and it is hard to read. 

Done.

Line 42: the reviewer suggests to indicate the duration of the follow-up from ref [3].

Done.

 Methods:

The control group is definitely too small in order to fully represent 'normality' in cognitive, neuropsychological and psychopathological domains in an age ranging from 6 and 9 years. This vulnus deserve to be better addressed in the limitations paragraph.  

We agree with the reviewer, the following sentence has been added to the limitations section:

“In particular, the sample of control children is too small to fully represent normality in cognitive, neuropsychological, or psychopathological domains, especially if one considers that the age group presents great interindividual variability.”

line 91: do the authors mean at least 36 week of age?

Yes. We have corrected the expression: “gestational age at birth ≥ 36 weeks”.

As all the children fluently spoke the Italian language it needs to be clarified in the text if all the tests performed (line 121-130) were actually validated in their nativa language.

All the tests used are standardized for the Italian population. The following sentence has been added: “All tests were corrected for age using published normative values, standardized for the Italian population.”

Results:

In the reviewer opinion psychopathological results should be removed for two reasons:

1)comparison lacks in statistical significance due to the even smaller  control group sample;

2)results are based on parental questionnaires. Their subjective point of view should not be taken into account in building a prognostic/pathological tool.

The reviewer believes all the psychopathological analysis should then be removed from the main text and submitted as a supplementary material.

The discussion section need to be restructured accordingly in light of the revision.

We partially agree with the reviewer. A smaller size participated in this second phase, but the reduction from the original group is less than 10% (the statistic reaches the statistical significance). Furthermore, with all the cautions of the case, the parental questionnaires are the most suitable tool and the most used in the literature in the age range studied.

We consider the part of psychopathology the most interesting and also only the unique that was not previously explored. We have added all the considerations of the reviewer in the limitation section, including the fact that these are only insights that need to be explored further with subsequent research.

“Although we did not explore the psychopathology extensively and our results are only preliminary, we found clinically relevant psychopathological symptoms in 35% of patients with HIE.

Regarding the in-depth assessment of executive functions, our patients are also too young for a complete evaluation of psychopathology. Our data are based on parents' reports, but it is known that reports about psychological symptoms may differ significantly between children and parents. As children reach an older age, we believe it will be necessary to perform a complete psychopathological examination, consisting of both the administration of questionnaires and a clinical interview performed by a trained psychotherapist ”.

A particular attention should be devoted to the limitations paragraph. The reviewer believes the sentence 'Therefore, our healthy group reflects general population trends and the real occurrence of cognitive problems in children without known risk factors' (line 270-271) cannot be stated with such a limited control group.

The sentence has been softened by the addition of the following:

“In particular, the sample of control children is too small to fully represent normality in cognitive, neuropsychological, or psychopathological domains, especially if one considers that the age group presents great interindividual variability. However, to reduce possible bias, healthy children were not recruited based on the absence of current neuropsychological dysfunctions but on the absence of preperinatal insults. Therefore, our healthy group reflects general population trends and the real occurrence of cognitive problems in children without known risk factors;..”

The results proposed, although interesting, are preliminary and need to the be extensively investigated on higher numbers and challenged with a pathological group secondary to different causes than HIE.

We agree with the reviewer, and we thank him for the interesting idea to compare the HIE patients with another pathological group. We have been added the following sentence to the limitations section:

“We hope to replicate the study with a greater number of patients, perhaps comparing them to a group of children with another pathological condition. This process could better highlight some specificities of HIE.”

Reviewer 2 Report

The article presented for review concerns a very important issue and is very interesting. The authors of the work pointed out the gaps in research in the analyzed area and they provide new and valuable information with their work. The number of methods used, which examine the functioning of the surveyed children in a multidimensional way is very impressing. All research techniques have been selected correctly, are adapted to the examination of children and give a reliable picture of their functioning. The methods of statistical analysis are clearly described, well chosen. The use of Boruta's algorithm is particularly interesting, as it allowed for the identification of IQ as the most important variable in the set of variables used. This made the analysis clear and easy to understand. The limitations of work are cleary presented.

I have a few comments:

There is a bit of a lack of a broader discussion of the results (some differences in results are statisticaly significant) presented in "Table 1. Groups characteristics by psychological tests." Such a broader discussion of the results would give an interesting picture of the functioning of the examined children. 

Among the mentioned research techniques used in the study, there are also several intended for the study of parents (lines 133 - 134). There is no discussion of the results of data analyzes from these techniques, and no relations with the results of other research techniques used in the study of children. 

In line 206 the authors say: "As hypothesized, we found that [...]" I miss a clearly stated hypothesis in the text, which should be placed, for example, at the end of the Introduction section.

The topic included in the Discussion in lines 243 - 256 is very interesting. Do the authors also connect it with the results of their research, especially with the intellectual functioning of the studied children? 

Author Response

Reviewer 2

The article presented for review concerns a very important issue and is very interesting. The authors of the work pointed out the gaps in research in the analyzed area and they provide new and valuable information with their work. The number of methods used, which examine the functioning of the surveyed children in a multidimensional way is very impressing. All research techniques have been selected correctly, are adapted to the examination of children and give a reliable picture of their functioning. The methods of statistical analysis are clearly described, well chosen. The use of Boruta's algorithm is particularly interesting, as it allowed for the identification of IQ as the most important variable in the set of variables used. This made the analysis clear and easy to understand. The limitations of work are cleary presented.

I have a few comments:

There is a bit of a lack of a broader discussion of the results (some differences in results are statisticaly significant) presented in "Table 1. Groups characteristics by psychological tests." Such a broader discussion of the results would give an interesting picture of the functioning of the examined children. 

We agree with the reviewer. We added a part in the discussion with a broader explanation of the results:

“Furthermore, we found differences in tasks evaluating visuomotor skills (Visual-Motor Integration Test and coding test). These abilities are typically impaired in children with a history of another pathological condition of the preperinatal period: preterm birth [25]. Despite the clinical differences between the disorders, mild-to-moderate conditions share common neurocognitive impairments that are considered disturbances of connectivity and are closely correlated with the integrity of the white matter [26].”

Among the mentioned research techniques used in the study, there are also several intended for the study of parents (lines 133 - 134). There is no discussion of the results of data analyzes from these techniques, and no relations with the results of other research techniques used in the study of children. 

Given the young age of the participants, we chose to evaluate the psychopathological profile only preliminarily by administering questionnaires to the parents. We hope, in the future, to be able to perform a complete psychopathological examination, consisting of both the administration of questionnaires and a clinical interview performed by a trained psychotherapist. Given this simple exploration of the psychopathological profile, we did not deepen the results, but we only calculated an index of psychopathological dysfunctions from clinical elevations of single psychological domains. This methodology provided dichotomous values and it is useful to quantify a wide range of dysfunctions in a unitary measure.

In line 206 the authors say: "As hypothesized, we found that [...]" I miss a clearly stated hypothesis in the text, which should be placed, for example, at the end of the Introduction section.

We agree with the reviewer; we added the hypothesis at the end of the introduction:

“We hypothesize that despite the neuroprotection offered by TH, the early stressful condition could increase children’s risk of developing a wide range of neurodevelopmental disorders and subtle diseases, such as neuropsychological or learning impairments. Early unfavorable events may trigger pro-inflammatory processes followed by a cascade of worsening events, which interact with multiple factors characterizing later life to deter-mine various complex scenarios.

The topic included in the Discussion in lines 243 - 256 is very interesting. Do the authors also connect it with the results of their research, especially with the intellectual functioning of the studied children? 

We tanks the reviewer for his comment. We have been added the following sentence to better link the topic to our results:

“In this framework, cognitive and psychopathological disorders among our patients could be considered two different manifestations of the same underlying neurobiological vulnerability.”

Round 2

Reviewer 1 Report

The authors revised the introduction section which is now easier to the reader. The reviewer does not get what the authors are stating at lines 67-68 as early (hours to days after HIE) biomarkers cannot really be useful in predicting long-term (years) outcomes.

The hypothesis formulated at the end of the introduction section (lines 77-79) is redundant and too vague. The reviewer believes it should be removed.

A few sentence still need a rephrasing as they generate a bit of confusion:

  • Lines 46-49;
  • Lines 55-58;
  • Lines 68-71.

Results

The reviewer has read with great interest the authors' rebuttal regarding psychopatological evaluation. The first critique did not regard the 10% lack in the HIE group but the (circa) 25% in the control group: this is a weakness that should, at least, be highlighted in the limitations. Since the authors state in rebuttal that a statistical significance has been reached the reviewer would like to ask how do the authors explain the high rate of false positives and the poor overall accuracy.

Although is clear that results are preliminary and the parents questionnaires are state-of-the-art in the age range investigated the authors -fairly- also state that patients at that age are too young for a complete, direct, psychopathological evaluation. Taken together these issues lead to the conclusion that perhaps the age range studied is too young to be correctly evaluated under a psychopathological point of view. Or, in other words, that parental point-of-view a that age would need to be corroborated longitudinally when children are old enough to directly undergo a psychopathological investigation.

Author Response

The authors revised the introduction section, which is now easier to the reader. The reviewer does not get what the authors are stating at lines 67-68 as early (hours to days after HIE) biomarkers cannot really be useful in predicting long-term (years) outcomes.

We agree with the reviewer; we reformulated the paragraph to advance clarity.

“A clear understanding of the incidence and nature of impairments after HIE treatment with TH could increase clinicians’ and researchers’ attention to long-term follow-ups and the search of early biomarkers of risk. This can allow the identification and monitoring of the most vulnerable children and provide timely interventions if necessary. During the time elapsed between the insult and disclosure of impairments, the developmental window for therapeutic interventions may be lost. In early infancy, rehabilitation programs can still favor changes in brain circuitry when neural correlates underlying more complex adaptive abilities can still occur. So, clear guidelines on follow-up and early biomarkers could be crucial to increasing the quality of life of children with HIE and reduce the burden on the mental health care system.”

The hypothesis formulated at the end of the introduction section (lines 77-79) is redundant and too vague. The reviewer believes it should be removed.

We added the hypothesis following the precise request of the other reviewer, so we think we cannot remove it. However, we have reduced it and made it more streamlined:

“We compare the neurodevelopment of a group of children with HIE treated with TH to that of a control group of healthy peers, evaluating cognitive, neuropsychological, and psychopathological criteria. We hypothesize that despite the neuroprotection offered by TH, the early stressful condition could increase children’s risk of developing a wide range of neurodevelopmental disorders and subtle diseases, such as neuropsychological or learning impairments.”

A few sentence still need a rephrasing as they generate a bit of confusion:

The sentences have been rephrased as follows:

  • Lines 46-49;

“Furthermore, group comparisons allow us to highlight subclinical vulnerabilities; that is, some borderline performances underlying weaknesses that could remain unchanged or worsen over time as more complex abilities emerge and the cumulative effects of several risk factors act synergistically.”

  • Lines 55-58;

“Given the established high incidence of long-term cognitive impairments before the era of TH usage, this result suggests a comparable incidence of neuropsychological difficulties despite TH treatment.”

  • Lines 68-71.

“A clear understanding of the incidence and nature of impairments after HIE treat-ment with TH could increase clinicians’ and researchers’ attention to long-term fol-low-ups and the search of early biomarkers of risk. This can allow the identification and monitoring of the most vulnerable children and provide timely interventions if necessary. During the time elapsed between the insult and disclosure of impairments, the develop-mental window for therapeutic interventions may be lost. In early infancy, rehabilitation programs can still favor changes in brain circuitry when neural correlates underlying more complex adaptive abilities can still occur. So, clear guidelines on follow-up and early biomarkers could be crucial to increasing the quality of life of children with HIE and re-duce the burden on the mental health care system.”

Results

The reviewer has read with great interest the authors' rebuttal regarding psychopatological evaluation. The first critique did not regard the 10% lack in the HIE group but the (circa) 25% in the control group: this is a weakness that should, at least, be highlighted in the limitations.

Yes, sorry, this was a mistake. We added in the limitations as follows:

“Furthermore, we obtained the scores of the questionnaires not from all parents of the sample; in particular, control’s parents reports lacking are about 24% of the entire sample of controls.”

 Since the authors state in rebuttal that a statistical significance has been reached the reviewer would like to ask how do the authors explain the high rate of false positives and the poor overall accuracy.

We think that simply having suffered from HIE cannot be considered a sentence of future psychopathology; many children recover and get well. This is the reason for the high incidence of false positives. We think that a better prevision could be obtained with early biomarkers, which could really indicate what children, among those suffering from HIE, are at risk, just to reduce false positives. We were, however, worried about the true positives that were increased. The overall accuracy is poor but still statistically significant. This could be due to the small sample size or could indicate that our hypothesis is wrong. Our data are only preliminary; as stated, a reevaluation with better tools and a larger sample is fundamental.

Although is clear that results are preliminary and the parents questionnaires are state-of-the-art in the age range investigated the authors -fairly- also state that patients at that age are too young for a complete, direct, psychopathological evaluation. Taken together these issues lead to the conclusion that perhaps the age range studied is too young to be correctly evaluated under a psychopathological point of view. Or, in other words, that parental point-of-view a that age would need to be corroborated longitudinally when children are old enough to directly undergo a psychopathological investigation.

We agree with the reviewer. Our attention to psychopathology comes from the fact that at the follow-up, the parent’s complaints about their son's behavior or psychological issues were very common. At that moment, we understood the real impact of this condition on the quality of life. So, even if early, we think that some signs of future psychopathology are already present in childhood and that it is not good to ignore them. However, the reviewer is definitely right when he says that they have to be followed longitudinally; we added the concept to the discussion:

“We think that these results have to be corroborated by following the children longitudinally. As children reach an older age, we believe it will be necessary to perform complete psychopathological examinations consisting of questionnaires and a clinical interview performed by a trained psychotherapist.”